# Revisiting the "satisfaction of spatial restraints" approach of MODELLER for protein homology modeling

Giacomo Janson[1]*, Alessandro Grottesi[2], Marco Pietrosanto[3], Gabriele Ausiello[3], Giulia Guarguaglini[4☯], Alessandro Paiardini[1☯]*

1 Department of Biochemical Sciences "A. Rossi Fanelli", "Sapienza" University of Rome, Roma, Italy, 2 Super Computing Applications and Innovation (CINECA), Roma, Italy, 3 Centre for Molecular Bioinformatics, Department of Biology, University of Rome Tor Vergata, Roma, Italy, 4 Department of Biology and Biotechnology, Institute of Molecular Biology and Pathology, "Sapienza" University of Rome, Roma, Italy

☯ These authors contributed equally to this work.
* giacomo.janson@uniroma1.it (GJ); alessandro.paiardini@uniroma1.it (AP)

## Abstract

The most frequently used approach for protein structure prediction is currently homology modeling. The 3D model building phase of this methodology is critical for obtaining an accurate and biologically useful prediction. The most widely employed tool to perform this task is MODELLER. This program implements the "modeling by satisfaction of spatial restraints" strategy and its core algorithm has not been altered significantly since the early 1990s. In this work, we have explored the idea of modifying MODELLER with two effective, yet computationally light strategies to improve its 3D modeling performance. Firstly, we have investigated how the level of accuracy in the estimation of structural variability between a target protein and its templates in the form of $\sigma$ values profoundly influences 3D modeling. We show that the $\sigma$ values produced by MODELLER are on average weakly correlated to the true level of structural divergence between target-template pairs and that increasing this correlation greatly improves the program's predictions, especially in multiple-template modeling. Secondly, we have inquired into how the incorporation of statistical potential terms (such as the DOPE potential) in the MODELLER's objective function impacts positively 3D modeling quality by providing a small but consistent improvement in metrics such as GDT-HA and lDDT and a large increase in stereochemical quality. Python modules to harness this second strategy are freely available at https://github.com/pymodproject/altmod. In summary, we show that there is a large room for improving MODELLER in terms of 3D modeling quality and we propose strategies that could be pursued in order to further increase its performance.

## Author summary

Proteins are fundamental biological molecules that carry out countless activities in living beings. Since the function of proteins is dictated by their three-dimensional atomic

**Data Availability Statement:** All relevant data are within the manuscript and its Supporting Information files. The data underlying the results presented in the study are also available from

https://github.com/pymodproject/altmod/tree/master/data.

**Funding:** GJ and AP received support from Associazione Italiana Ricerca sul Cancro (AIRC, https://www.airc.it/) MFAG 20447 and Progetti Ateneo Sapienza University of Rome (https://www.uniroma1.it). GG received support from Associazione Italiana Ricerca sul Cancro (AIRC, https://www.airc.it/) IG Grant 17390. AP, AG and GJ acknowledge the CINECA award under the ISCRA initiative, for the availability of high performance computing resources and support (IsC68_altmod). The funders had no role in study design, data collection and analysis, decision to publish, or preparation of the manuscript.

**Competing interests:** The authors have declared that no competing interests exist.

structures, acquiring structural details of proteins provides deep insights into their function. Currently, the most frequently used computational approach for protein structure prediction is template-based modeling. In this approach, a target protein is modeled using the experimentally-derived structural information of a template protein assumed to have a similar structure to the target. MODELLER is the most frequently used program for template-based 3D model building. Despite its success, its predictions are not always accurate enough to be useful in Biomedical Research. Here, we show that it is possible to greatly increase the performance of MODELLER by modifying two aspects of its algorithm. First, we demonstrate that providing the program with accurate estimations of local target-template structural divergence greatly increases the quality of its predictions. Additionally, we show that modifying MODELLER's scoring function with statistical potential energetic terms also helps to improve modeling quality. This work will be useful in future research, since it reports practical strategies to improve the performance of this core tool in Structural Bioinformatics.

## Introduction

*In silico* protein structure prediction constitutes an invaluable tool in Biomedical Research, since it allows to obtain structural information on a large number of proteins currently lacking an experimentally-determined 3D structure [1]. Template-based modeling (TBM) is the most frequently employed prediction strategy. In the past years it has been considered as the most accurate one [2], but recently it has been shown that template-free strategies have reached comparable levels of performance with protein targets that lack good templates [3] (for example, with members of several membrane protein families [4]). Despite this fact, TBM methods, thanks to their speed, flexibility and growing template libraries [5], currently remain the instrument of choice for many researchers.

Homology modeling (HM) is a fast and reliable TBM method in which a target protein is modeled by using as a structural template an homologous protein. HM predictions usually consist of three phases. In the first, the sequence of the target is used to search for suitable templates in the PDB [6–7]. In the second, a sequence alignment between the target and templates is built with the goal of inferring the equivalences between their residues [8]. In the final, the information of the templates is used to build a 3D atomic model of the target.

The overall accuracy of HM has remarkably increased in the last 25 years. While a major factor for this advancement has been the increase of the size of sequence and structural databases [5], it has been shown that progress in HM algorithms has also played a key role [9]. These improvements have consisted mainly in advances in template searching and alignment building algorithms, while only minor advances have been witnessed in the 3D model building step [10]. However, recent breakthroughs in protein structure refinement methods [11–12] envisage a large room for improvement in HM which could originate from advances in 3D model building.

MODELLER [13] is the most frequently used program for 3D model building in HM. One of the main reasons of its success has been its accurate [14], yet fast algorithm. In MODELLER, the information contained in an input target-template alignment is used to generate a series of homology-derived spatial restraints (HDSRs), acting on the atoms of the 3D protein model. Sigma ("$\sigma$") values of homology-derived distance restraints (HDDRs) determine the amount of conformational freedom which the model is allowed to have with respect to its templates. MODELLER uses a statistical "histogram-based" strategy to estimate $\sigma$ values [15]. These restraints are incorporated into an objective function which also includes physical energetic

terms from CHARMM22 [16]. A fast, but effective optimization algorithm based on a combination of conjugate gradients (CG) and molecular dynamics with simulated annealing (MDSA) is then used to identify a model conformation that satisfies as much as possible the HDSRs, while retaining stereochemical realism.

The core MODELLER algorithm was developed in the early 1990s and it was essentially left unchanged over the years. Despite its importance, there have been relatively few attempts to improve it.

In 2015, Meier and Söding designed a novel probabilistic framework for building HDDRs [10], whose aim was to help MODELLER tolerate alignment errors and to combine the information from multiple templates in a statistically rigorous way. This system increased 3D modeling quality, especially for multiple-template modeling. However, since it is integrated in the HHsuite project [17] it can be employed only when the first two phases of HM are carried out by programs of the HHsuite package.

Researchers from Lee's group developed a modified version of MODELLER which they have been using in CASP experiments [18–20]. First, they replaced the MODELLER optimization algorithm with the more thorough conformational space annealing (CSA) method [21]. Secondly, they pioneered a new strategy to assign $\sigma$ values to HDDRs relying on machine learning [22]. Finally, they included a series of additional terms to the MODELLER objective function, such as terms for the DFIRE [23] and DFA [24] knowledge-based potentials, for hydrogen bond formation [25] and to enforce in models predictions of structural properties. In terms of 3D modeling quality, this system outperformed the original MODELLER [20]. Unfortunately, the separated contribution of several of these modifications is not reported and much of this system remains in-house (only the CSA algorithm is publicly available).

Although these seminal studies have shown that the core MODELLER algorithm has room for improvement, most of its users employ its original version, probably because existing modifications either depend on additional packages to install, or are computationally too expensive (e.g., the CSA algorithm alone was reported to increase computational times by a factor of ~130). Since MODELLER is a core tool in Structural Bioinformatics, it is of paramount importance to investigate in detail the inner working of its algorithm and to develop it further. Here, we have explored two computationally light strategies to improve it in terms of 3D modeling quality.

Particular attention has been dedicated in understanding how the level of accuracy in the estimation of structural variability between the target and templates expressed as $\sigma$ values influences 3D modeling. Although in this work we have not modified the MODELLER algorithm for $\sigma$ values assignment, we propose strategies that could be likely pursued in the next-future in order to greatly increase the performance of the program. Additionally, we have investigated how the incorporation of statistical potential terms, such as DOPE [26], in the program's objective function is able to impact positively 3D modeling and under certain conditions (for example in single-template modeling) it can be coupled synergistically to the previous strategy.

To rigorously validate these approaches, we have benchmarked them using protein targets from a diverse set of high-resolution structures from the PDB and we quantified the individual impact on 3D modeling of each modification. This information will be useful in future research, since it shows in which areas there is still room for improvement and in which areas it might be difficult to advance further.

## Materials and methods

### Outline of MODELLER's homology-derived distance restraints

The MODELLER approach relies on the generation of HDSRs for interatomic distances and dihedral angles [15]. Each HDSR is treated as a probability density function (*pdf*). HDSRs

acting on interatomic distances (that is, HDDRs) have a predominant role in determining the 3D structure of a model. The way they are built is summarized here.

For a couple of atoms $i$ and $j$ of the model, the program finds in the template the equivalent atoms $k$ and $l$ which have a distance in space of $d_t$. The distance $d_m$ between $i$ and $j$ is assumed to be normally distributed around $d_t$ with a standard deviation $\sigma$ and the *pdf* restraining it is:

$$f(d_m) = \frac{1}{\sigma\sqrt{2\pi}} e^{\frac{-(d_m - d_t)^2}{2\sigma^2}}. \tag{1}$$

In MODELLER *pdfs* are converted in objective function terms as follows:

$$obj(d_m) = -ln(f(d_m)) = -ln\left(\frac{1}{\sigma\sqrt{2\pi}} e^{\frac{-(d_m - d_t)^2}{2\sigma^2}}\right) = \frac{(d_m - d_t)^2}{2\sigma^2} - ln\left(\frac{1}{\sigma\sqrt{2\pi}}\right), \tag{2}$$

therefore Gaussian HDDRs correspond to harmonic potential terms. Since HDDRs are considered to be independent, their objective function terms are summed. HDDRs are built for four groups of atoms: the Cα-Cα, backbone NO, side chain-main chain (SCMC) and side chain-side chain (SCSC) groups (see **S2 Table**). MODELLER generates its $\sigma$ values (hereinafter named $\sigma_{MOD}$ values) through an histogram-based approach [15].

MODELLER allows to take advantage of multiple templates, a strategy that (when templates are chosen adequately) usually outperforms single-template modeling [27]. When employing $U$ templates to restrain a distance $d_m$, MODELLER uses the following *pdf*:

$$f(d_m) = \sum_{u=1}^{U} w_u \frac{1}{\sigma_u\sqrt{2\pi}} e^{\frac{-(d_m - d_{t,u})^2}{2\sigma_u^2}}, \tag{3}$$

where $u$ is the template index, $w_u$ is a template-specific weight, $d_{t,u}$ and $\sigma_u$ are the distance observed in template $u$ and its $\sigma$ value respectively. In MODELLER, $w_u$ is a function of the local sequence similarity between the target and template $u$.

The total objective function of MODELLER ($F_{TOT}$) can be expressed as follows:

$$F_{TOT} = F_{PHYS} + F_{HOM}, \tag{4}$$

where $F_{PHYS}$ contains five physical terms (see **S1 Table**) and $F_{HOM}$ contains HDSRs terms. In this work, the weights for $F_{PHYS}$ and $F_{HOM}$ were always left to 1.0 (therefore they are omitted from the formula above).

## Benchmarking MODELLER modifications with an analysis set

In order to benchmark modifications of MODELLER, we built an analysis set of selected target proteins. We obtained 926 X-ray structure chains from PISCES [28], using the following criteria to filter the PDB:

- the maximum mutual sequence identity (SeqId) among the chains was 10%;

- their structures had a resolution < 2.0 Å and R-factor < 0.25;

- they contained no missing residues due to lacking electron density;

- their length was between 70 and 700 residues.

These chains were our target candidates. To obtain their templates, we culled from PISCES another set using similar filters, except that this time the maximum mutual SeqId was 90%. We removed from this larger set all the targets, obtaining 6224 chains. Each target was then

aligned to these chains using TM-align [29] and we selected as template candidates the chains meeting the following criteria:

- the SeqId in the structural alignment built by TM-align was between 15% and 95%;

- the two TM-scores [30] produced by TM-align (each score is normalized by the length of one of the aligned proteins) were at least 0.6, a threshold to consider two proteins as homologous [31].

We retained for each target only its top five templates in terms of TM-score (normalized on the target length). In this way, we obtained a final set of 225 target chains (suitable templates could not be found for 701 targets, a result of using only high-resolution template structures). For each target, we performed single-template modeling only with its top template and therefore we had 225 single-template models, which constituted the Analysis Single-template (AS) set. 118 targets had at least two templates (with an average of 3.3), which constituted the Analysis Multiple-templates (AM) set.

The average SeqId for the AS target-template alignments is 0.38. Improving the performance of MODELLER with targets having templates with a SeqId < 0.40 is important, because these cases are the most frequent ones in Biomedical Research [5] and the accuracy of TBM is often low in this regimen. The well-equilibrated distributions of SeqId, target coverage, target length and of CATH structural classes [32] of the analysis set (see **S1 Fig**) assure that our results have a general validity.

### Alignment building

In order to align target-template pairs we employed the accurate HHalign program [7], which confronts two profile hidden Markov models. To build input profiles for HHalign, we ran HHblits [33] with its default parameters and three search iterations against the *uniprot20_2016_02* database. After employing HHalign to align pairs of target-template profiles, we extracted from the program's output their pairwise alignments. Multiple target-templates alignments were obtained by joining pairwise alignments.

Whenever specified, we also employed target-template alignments built with TM-align in order to assess the effect on 3D modeling of HDDRs derived from error-free structural alignments.

### 3D model building and evaluation

For all benchmarks we used MODELLER version 9.21. In order to modify its objective function terms and optimization schedules we interfaced with its Python API. To modify the restraints parameters we employed Python scripts to edit the default restraints files generated by the program (see the "Restraints files building" section).

In MODELLER, the final quality of a model is largely determined in the MDSA phase. In this work, unless otherwise stated, we employed the default *very_fast* MDSA protocol of the program (corresponding to a 5.4 ps run). When specified, we also employed the more thorough *slow* protocol (corresponding to a 18.4 ps run). The CG protocol was always left to its default parameters.

The approach used to evaluate the quality of an homology model was to build 16 different copies of it (hereinafter defined as decoys), and to report as an overall quality score (see below) the average score of the 16 decoys.

To evaluate the quality of the backbones we used the GDT-HA metric [9] computed by the TM-score program. In order to evaluate the quality of local structures and side chains, we used the lDDT metric [34], computed by the lDDT program. Detailed descriptions of these two

metrics are given in **S1 Text**. To evaluate the stereochemical quality of models we employed MolProbity scores computed by the MolProbity suite [35]. A MolProbity score expresses the global stereochemical quality of a 3D model. The lower it is, the higher is the quality of the model.

## Optimal $\sigma$ values for homology-derived distance restraints

$\sigma$ values of HDDRs have a fundamental role in MODELLER. A natural question is: given a target-template alignment, what is the set of $\sigma$ values which will maximize 3D modeling accuracy? The concept of optimal $\sigma$ values in single-template modeling was addressed for the first time by the Lee group [22]. They reported that for a Gaussian HDDR acting on a distance $d_m$ between atoms $i$ and $j$ in a 3D model, the optimal $\sigma$ value is:

$$|\Delta d_n| = |d_n - d_t|, \tag{5}$$

where $d_t$ is the distance between the template atoms equivalent to $i$ and $j$ and $d_n$ is the distance between $i$ and $j$ observed in the experimentally-determined native target structure. We show that the use of $|\Delta d_n|$ values for Gaussian HDDRs is supported by theory, as it can be analytically proven that they maximize the likelihood of obtaining a model in which each restrained $d_m$ is equal to its corresponding $d_n$ (see **S2 Text**).

In the case of multiple-template HDDRs, we demonstrate that the combination of optimal $\sigma$ values and weights can be found again analytically (see **S3 Text**). In this situation, the optimal $\sigma$ values are again $|\Delta d_n|$ values. The associated template weighting scheme assigns a weight of 0 to all templates with the exception of the template with the lowest $\sigma$, which should have a weight of 1. We termed this scheme as the "only-lowest" (OL) scheme. Note that the OL scheme is an extreme case of the weighting scheme proposed in [36] (see **S3 Text**).

Whenever using $|\Delta d_n|$ values as $\sigma$ parameters, we had to modify them by setting their minimum value at 0.05 Å. Raw $|\Delta d_n|$ values are extracted directly from pairs of homologous protein structures and they are often close to 0 Å (see **Fig 1A**). In MODELLER, HDDRs having very small $\sigma$ values will seldom be satisfied because their quadratic objective function terms will penalize enormously even minimal deviations from templates. In fact, using unmodified $|\Delta d_n|$ values often leads to modeling failures, since the total objective function of models surpasses the allowed limit of MODELLER, stopping the model building process. Setting a lower limit to their value, allows their use in 3D modeling.

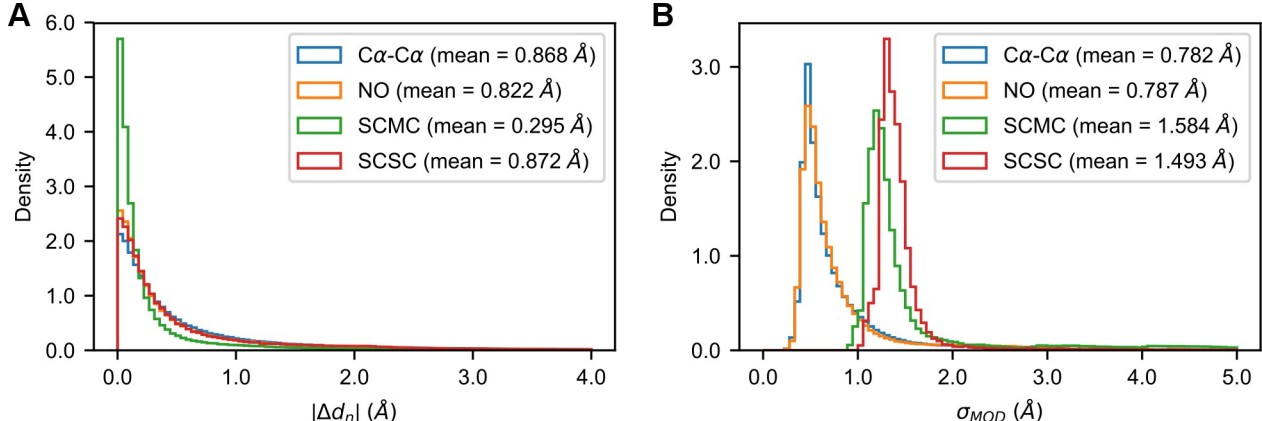

**Fig 1. Distribution of $|\Delta d_n|$ and $\sigma_{MOD}$ values.** Distributions of the $|\Delta d_n|$ (A) and $\sigma_{MOD}$ (B) values observed in the AS models for the four HDDR groups of MODELLER. Beside the names of the restraints groups, their mean values are reported.

## Restraints file building

In MODELLER, the restraints used in the 3D model building phase are supplied in a specific file. In this work, we explored how the choice of parameters for the HDDRs influences 3D modeling. Therefore, our approach for building restraints files was to let MODELLER generate its default restraints files and then to modify it by leaving unaltered all the stereochemical and homology-derived dihedral angle restraints and by modifying only the HDDRs parameters. The Python code we used to customize restraints files is available at https://github.com/pymodproject/altmod. This code allows to modify the list of HDDRs of a model by supplying a list of user-specified parameters for them. Users may specify both the location and scale parameters of the Gaussian HDDRs, see Eq (1). Additionally, we provide code for running MODELLER with optimal HDDRs, which can be employed whenever users can supply a native structure file for the target protein.

## Perturbing optimal $|\Delta d_n|$ values

To understand the effect of using error-containing $|\Delta d_n|$ estimations on 3D modeling, we randomly perturbed the $|\Delta d_n|$ values of target-template pairs. Our aim was to simulate a series of $|\Delta d_n|$ estimators having different performances in terms of the Pearson correlation coefficient (PCC) between the perturbed values and their unperturbed counterparts. To perturb a $|\Delta d_n|$ list of a target-template pair in order to reach a selected PCC (referred to as $PCC_{SEL}$), we added to the $\Delta d_n$ values of the pair a list $\varepsilon$ of errors in order to obtain a list $p$ of perturbed values, whose elements are:

$$p_i = |\Delta d_n + \varepsilon_i|, \tag{6}$$

where $p_i$ is the $i$-th perturbed value and $\varepsilon_i$ is a random error extracted from a Laplace distribution with location 0 and scale parameter $b$. We chose Laplace distributions since adding errors extracted from them results in $p_i$ values being distributed approximately as exponentials, which resemble the original $|\Delta d_n|$ distributions (see **Fig 1A** and also **Fig F** in **S2 Fig**). To obtain the desired $PCC_{SEL}$, a $|\Delta d_n|$ list was perturbed in 5000 trials by drawing $\varepsilon$ from Laplace distributions with different $b$ values and in each trial the PCCs between $p$ and the original $|\Delta d_n|$ list was computed ($b$ was linearly increased from $0.005^* m_{obs}$ to $25.0^* m_{obs}$, where $m_{obs}$ is the mean of the $|\Delta d_n|$ list). At the end of these trials, the $p$ list having the observed PCC as close as possible to $PCC_{SEL}$ was selected. The $|\Delta d_n|$ values of each HDDR group (that is, the Cα-Cα, NO, SCMC and SCSC groups) were perturbed independently. This heuristic procedure usually selects PCCs being not more than 0.003 units away from $PCC_{SEL}$ (thus giving practically the same level of perturbation of $PCC_{SEL}$). Note that larger $b$ values result in larger amounts of noise (and in lower PCCs), but they also increase the mean of $p$ lists far beyond the typical values observed for $|\Delta d_n|$ data. Therefore, the $p$ lists of the four HDDR groups were always scaled so that their mean (referred to as $m_{pt}$) would be equal to:

$$m_{pt} = (PCC_{SEL}) m_{obs} + (1 - PCC_{SEL}) m_{grp}, \tag{7}$$

where $m_{grp}$ is the mean $|\Delta d_n|$ value observed in all our AS models for HDDR group *grp* (see **Fig 1A**). In this way, if $PCC_{SEL}$ is near 1, the mean of a $p$ list tends to the mean of the unperturbed $|\Delta d_n|$ list, while if $PCC_{SEL}$ is near 0, its mean tends to the "global" mean observed for the corresponding group in our whole $|\Delta d_n|$ data sets. This whole perturbation scheme has three important characteristics. (i) Adding noise to every $|\Delta d_n|$ value of a target-template pair allows us to properly simulate the effect of a real-life $|\Delta d_n|$ estimator in which every estimation would have some uncertainty. (ii) Since 3D modeling quality tends to decrease when the average $\sigma$ value of a model increases (see **Fig 2A** and **2B**), the scaling procedure ensures that when employing

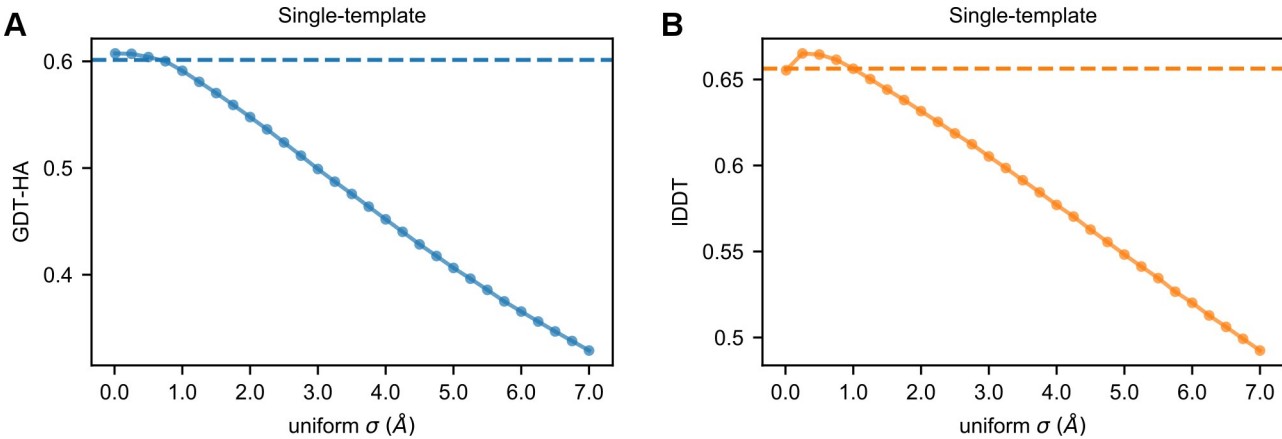

**Fig 2. Modeling with uniform σ values.** Average GDT-HA (A) and lDDT (B) scores of the AS models as a function of the uniform σ value (ranging from 0.01 to 7.0 Å) applied to their HDDRs. The horizontal dashed lines represent the average scores obtained with the original $\sigma_{MOD}$ values.

highly perturbed $|\Delta d_n|$ lists, alterations in the quality of 3D models will not be caused by just unrealistically increasing their mean σ. (iii) The choice of Laplace distributions as error-generators ensures that alterations in quality will not be caused by drastically changing the shape of the perturbed values distributions with respect to the observed $|\Delta d_n|$ ones. An example of the effects of this perturbation scheme are found in **S2 Fig** (while the code that implements it is available in our Git repository, see above).

To simulate various levels of accuracy in $|\Delta d_n|$ estimation, we used 10 $PCC_{SEL}$ values (linearly spacing from 0.0 to 0.9). For each $PCC_{SEL}$, we generated 5 sets of perturbed $|\Delta d_n|$ values per target-template pair, which allowed to better sample the effect of perturbations. For each perturbed set, we built 8 decoys per target with MODELLER (resulting in a total of $5^{*}8 = 40$ decoys per target for each $PCC_{SEL}$ value). For a certain $PCC_{SEL}$ value, the quality score for a 3D model was recorded as the average score of all its 40 decoys.

To quantify in terms of PCC the actual amount of perturbation introduced in the $|\Delta d_n|$ values of a single model, we used a score defined as $PCC_{MODEL}$. This score is computed as:

$$PCC_{MODEL} = \frac{1}{n_R}\sum_{r=1}^{n_R}\left(\frac{1}{U}\sum_{u=1}^{U}PCC_{u,r}\right),\tag{8}$$

where $n_R$ is the number of perturbed $|\Delta d_n|$ sets (in our case 5), $r$ is the index for these sets, $U$ is the number of templates of the model and $PCC_{u,r}$ indicates the observed PCC between the list of $|\Delta d_n|$ values associated with the $u$-th template and the corresponding list of perturbed values in set $r$. For each HDDR group, the relationship between $PCC_{SEL}$ values and the average $PCC_{MODEL}$ observed in the AS and AM sets is almost perfectly linear (see **Fig G** and **H** in **S2 Fig**), confirming the efficacy of the perturbation scheme.

## Inclusion of statistical potential terms in the objective function of MODELLER

In this work, we explored the effect of including in the objective function of MODELLER terms for interatomic distance statistical potentials. These potentials are developed with the aim of recognizing native-like protein conformations [37], therefore their use could help MODELLER to approach these conformations [38].

We employed the DOPE potential [26], which is integrated in the MODELLER package where it is commonly used to evaluate qualities of 3D models. DOPE is an "all atom" potential.

Its 12561 terms are approximated with interpolating cubic splines, which can be differentiated analytically and used in the gradient-based optimization algorithm of the program.

The Lee group previously included the DFIRE [23] potential in the MODELLER objective function [18]. To compare their performances in 3D model building, we also integrated DFIRE in MODELLER (DFIRE parameters were obtained from its source code).

When including statistical potential terms, the MODELLER objective function becomes:

$$F_{TOT} = F_{PHYS} + F_{HOM} + w_{SP}F_{SP}, \qquad (9)$$

where $F_{SP}$ contains the statistical potentials terms and $w_{SP}$ is their weight. For obtaining best 3D modeling results, we tested several values of $w_{SP}$.

We employed statistical potentials using a contact shell value of 8.0 Å. Higher values can be safely avoided because the terms of DOPE and DFIRE start to acquire a flat shape over the 8.0 Å threshold (see **Fig A** in **S3 Fig**). The code we used to employ these potentials in MODELLER is freely available at https://github.com/pymodproject/altmod.

## Results

### Effects of optimal $\sigma$ values on 3D modeling

**Effects on single-template modeling.**  Gaussian HDDRs are the heart of the MODELLER approach. At first, we explored how the use of optimal $\sigma$ values (that is, $|\Delta d_n|$ values) influences single-template modeling. The Lee group already reported it to bring significant improvements for a small number of proteins. Here, we extended the analysis to a larger set to derive general conclusions. As shown in **Table 1**, employing restraints bearing $|\Delta d_n|$ values greatly increases 3D modeling accuracy. In terms of global Cα backbone quality, the average GDT-HA score of the AS models increases by 6.0% with respect to the score obtained with $\sigma_{MOD}$ values. An improvement is also observed for local all-atom quality, as the average lDDT score increases by 4.2%. Increments in GDT-HA and lDDT are seen for 224/225 and 225/225 AS models respectively (see **Fig 3A** and **3B**).

Increasing target-template alignment quality is one of the current challenges in TBM. In our AS models, the average accuracy of HHalign sequence alignments with respect to error-free TM-align structural alignments is 0.87 (see **S4 Fig**). When rebuilding the AS models using

**Table 1. 3D modeling qualities of the AS single-template models built with optimal HDDRs and alignments.**

| Strategy | GDT-HA | lDDT | MolProbity score |
|---|---|---|---|
| MODELLER[a] | 0.6014 (-) | 0.6563 (-) | 3.0104 (-) |
| OPTIMAL[b] | 0.6377 (+6.0%)* | 0.6842 (+4.2%)* | 3.0311 (+0.7%)* |
| MODELLER-SLOW[c] | 0.6036 (+0.4%)* | 0.6594 (+0.5%)* | 2.8512 (-5.3%)* |
| OPTIMAL-SLOW | 0.6377 (+6.0%)* | 0.6853 (+4.4%)* | 2.9039 (-3.5%)* |
| MODELLER-TMalign[d] | 0.6383 (+6.1%)* | 0.6951 (+5.9%)* | 3.0411 (+1.0%)* |
| OPTIMAL-TMalign | 0.6805 (+13.2%)* | 0.7259 (+10.6%)* | 3.0870 (+2.5%)* |

The "GDT-HA", "lDDT" and "MolProbity score" columns report the average values for those metrics. Percent improvements are computed with respect to the scores of the default MODELLER (first row).

[a]The "MODELLER" prefix indicates that the strategy employs HDDRs generated by MODELLER.

[b]The "OPTIMAL" prefix indicates the use of optimal HDDRs.

[c]The "SLOW" suffix indicates the use of the *slow* MDSA protocol instead of the default *very_fast* one.

[d]The "TMalign" prefix indicates the use of target-template alignment built through TM-align.

*Asterisks denote a statistically significant difference (according to a Wilcoxon signed-rank test with a significance level of 0.05) between the scores of a strategy and the scores of the default MODELLER. See **S3 Table** for a full list of the numerical p-values.

$\sigma_{MOD}$ values and TM-align alignments, the average GDT-HA and lDDT scores improve by 6.1% and 5.9% respectively over the scores obtained with $\sigma_{MOD}$ values and HHalign alignments (see **Table 1**). These results show that by optimizing parameters of the 3D model building phase of single-template HM, the same improvement obtainable by optimizing alignment building can be reached.

It might be thought that $|\Delta d_n|$ values aid 3D modeling by compensating for alignment errors, that is, by assigning misaligned residues more conformational freedom to help MODELLER repositioning them in a correct way. However, their effect can not be explained only by this mechanism, since they yield a 6.6% and 4.4% increase in GDT-HA and lDDT also when models are built with TM-align alignments (see **Table 1**).

**Effects on multiple-template modeling.** Next, we explored the effect of optimal HDDRs in multiple-template modeling, which has never been assessed before. As shown in **Table 2**, applying an optimal set of $\sigma$ values and template weights results in an enormous improvement in the quality of 3D models (see also **Fig 3C and 3D**). When building the AM models with optimal restraints, their average GDT-HA and lDDT scores improve by 38.9% and 18.9% over the scores obtained by using MODELLER-generated restraints. These increments are larger than the one observed when performing multiple-template modeling with MODELLER-generated restraints and error-free TM-align structural alignments, which results in a 5.7% and 5.1% improvements in GDT-HA and lDDT.

Optimal HDDRs increase even more the beneficial effect of using multiple templates. With MODELLER-generated restraints, employing multiple templates leads to an improvement of 1.9% and 2.0% in the average GDT-HA and lDDT of the AM models over single-template modeling performed with top-templates (see the MODELLER-ST strategy in **Table 2**). On the other hand, with optimal HDDRs, it leads to an improvement of 33.2% and 16.0% in GDT-HA and lDDT over single-template modeling performed with optimal HDDRs (see the OPTIMAL-ST strategy in **Table 2**).

The reason for this large improvement is the following. In MODELLER, the *pdf* for a multiple-template HDDR includes a weighted contribution from each template. In optimal HDDRs, $|\Delta d_n|$ values are employed as $\sigma$ values in conjunction with the OL weighting scheme (see the "Methods" section). In this scheme, only the contribution of the best template is selected for each HDDR (when considering a single HDDR, the best template is defined as the one having a distance $d_t$ as close as possible to the target distance $d_n$, that is, the template with lowest $|\Delta d_n|$ value). On the other hand, in MODELLER-generated HDDRs, the weights are usually non-zero for every template, meaning that the contribution of the best template is always weakened. This effect increases the allowed conformational space for the restrained distance, thus making it less likely to build a model with a near-native distance.

The importance of the template-weighting scheme [10] is illustrated by the fact that when employing $|\Delta d_n|$ values and a uniform weighting scheme (that is, for an HDDR with $U$ templates each template is given a weight $w_u = 1/U$), the average GDT-HA and lDDT scores of the AM models improve only by 18.3% and 8.9% over the standard MODELLER (see the OPTIMAL-U strategy in **Table 2**).

Our data shows that if the best template can be identified for each restrained distance, a substantial improvement in 3D modeling quality can be reached. A relevant matter is therefore to understand whether for a single residue (on which several HDDRs are usually acting) or for some stretch of contiguous residues, the best template always happens to be the same, or instead if multiple templates are effectively used together. **S4 Text** shows that in the AM set, for regions of the target sequences covered by multiple templates, all templates are frequently used at the same time. When every template has the same level of sequence similarity with the target (e.g.: all of them have around 30% SeqId), usually no template dominates over some

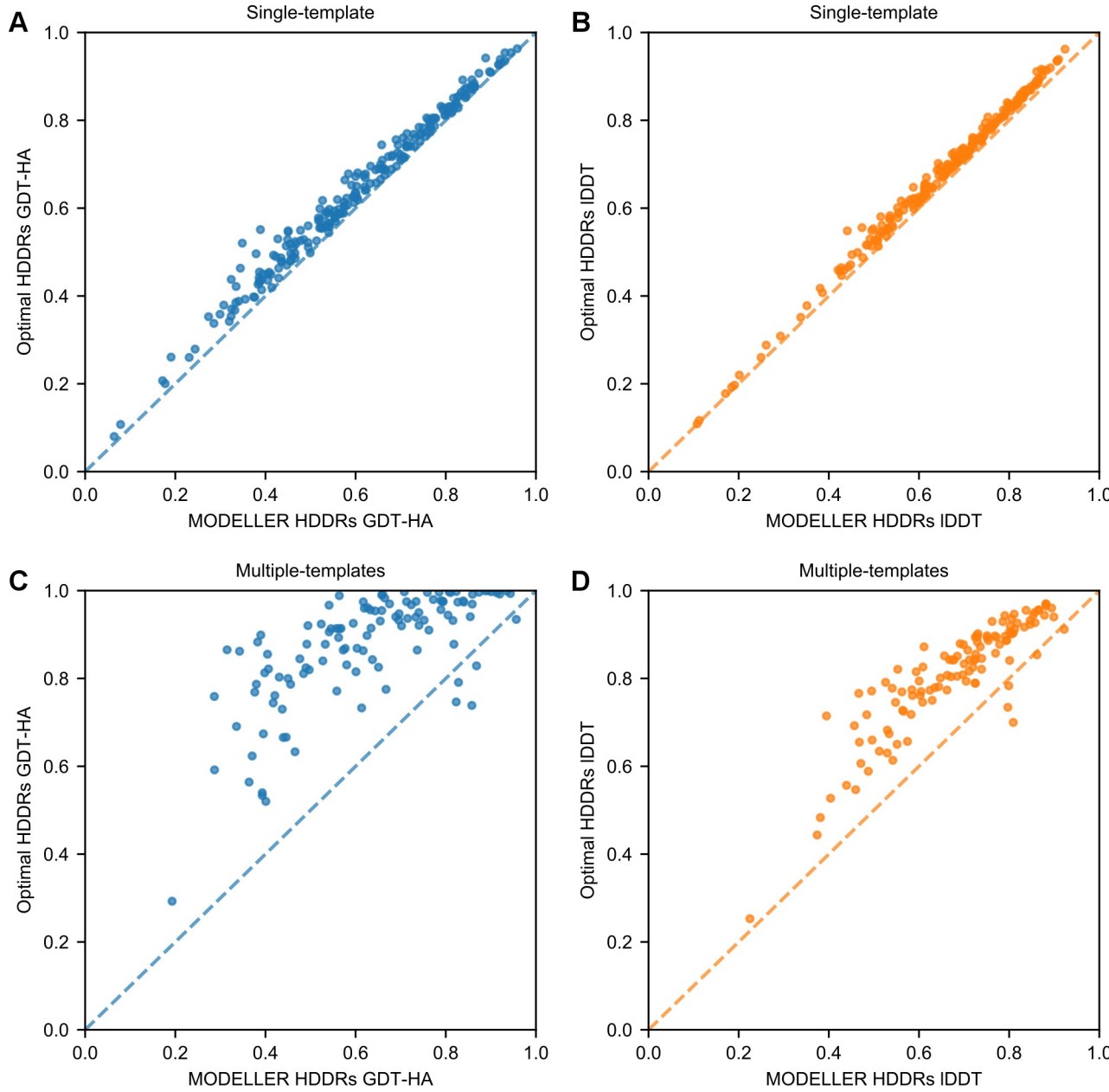

**Fig 3. The use of optimal parameters for HDDRs improves 3D modeling quality.** (A) and (B) GDT-HA and lDDT scores of the AS models built with $\sigma_{MOD}$ (reported on the x-axis) and with optimal $|\Delta d_n|$ (y-axis) values. (C) and (D) GDT-HA and lDDT scores for the AM models obtained with MODELLER-generated (x-axis) and optimal (y-axis) HDDRs.

extended region of the target sequence. Even when considering only the HDDRs acting on a single residue, different best templates are most often used concomitantly (see **Fig A** and **C** in **S4 Text**). In situations where there is a large difference in SeqId among the templates, even if the template with the highest SeqId tends to be picked more frequently, also other templates may maintain a significant contribution throughout the target sequence (see **Fig B** in **S4 Text**). Therefore, in order to optimally model most target residues, multiple templates have to be effectively used. This fact suggests that in order to harness the full potential of multiple-templates in homology modeling, the concept of the "best template" for single restrained distances

**Table 2. 3D modeling qualities of the AM multiple-template models built with optimal HDDRs and alignments.**

| Strategy | GDT-HA | lDDT | MolProbity score |
|---|---|---|---|
| MODELLER | 0.6287 (-) | 0.6819 (-) | 3.0725 (-) |
| OPTIMAL | 0.8733 (+38.9%)* | 0.8106 (+18.9%)* | 3.1478 (+2.4%) |
| MODELLER-SLOW | 0.6310 (+0.4%)* | 0.6850 (+0.5%)* | 2.9143 (-5.2%)* |
| OPTIMAL-SLOW | 0.8747 (+39.1%)* | 0.8133 (+19.3%)* | 3.0475 (-0.8%)* |
| OPTIMAL-U[a] | 0.7438 (+18.3%)* | 0.7427 (+8.9%)* | 3.1744 (+3.3%) |
| MODELLER-ST[b] | 0.6168 (-1.9%)* | 0.6683 (-2.0%)* | 3.0231 (-1.6%)* |
| OPTIMAL-ST | 0.6557 (+4.3%)* | 0.6986 (+2.5%)* | 3.0398 (-1.1%) |
| MODELLER-TMalign | 0.6645 (+5.7%)* | 0.7165 (+5.1%)* | 3.0529 (-0.6%) |
| OPTIMAL-TMalign | 0.9222 (+46.7%)* | 0.8498 (+24.6%)* | 3.1044 (+1.0%) |

See **Table 1** for the description of contents, columns and most modeling strategies names.

[a]The "U" suffix indicates the use of uniform template weights for multiple-template HDDRs.

[b]The "ST" suffix indicates that only the top template for each target was used (thus resulting in single-template modeling).

*Asterisks denote a statistically significant difference (according to a Wilcoxon signed-rank test with a significance level of 0.05) between the scores of a strategy and the scores of the default MODELLER. See **S4 Table** for a full list of the numerical p-values.

should be considered. In a real-life protein structure prediction scenario (where the structure of the target protein is unknown), the ability to select the best template for each distance would be related to our ability to directly estimate $|\Delta d_n|$ values. If our accuracy in estimation is sufficiently high, the impact on multiple-template 3D modeling quality will be largely beneficial (see the "Perturbing optimal σ values" section).

**Effects on stereochemical quality.** In both single and multiple-template modeling, the use of optimal HDDRs appears to decrease the stereochemical quality of models, as seen by increased MolProbity scores (see **Table 1** and **Table 2**). The increment is more prominent in multiple-template modeling (2.4%) than in single-template modeling (0.7%). While optimal restraints may guide the models in conformations near the native state, at the same time they probably force stereochemical inaccuracies. However, employing a more through MDSA

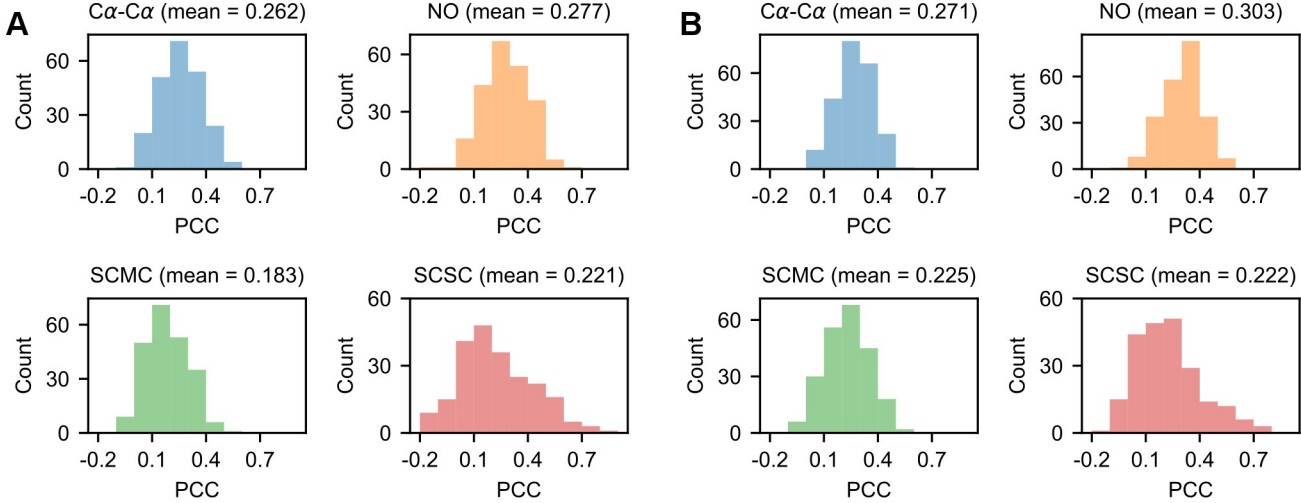

**Fig 4. Correlation between $\sigma_{MOD}$ and $|\Delta d_n|$ values in the AS models.** (A) Distributions for the PCCs between $\sigma_{MOD}$ and $|\Delta d_n|$ values for the HDDRs of the 225 AS models. (B) PCC distributions for the AS models rebuilt with TM-align alignments.

protocol is sufficient to almost entirely relax these inaccuracies, while maintaining high GDT-HA and lDDT scores (see the strategies with the "SLOW" suffix in the tables).

### Perturbing optimal $\sigma$ values

As first demonstrated in [22], $\sigma_{MOD}$ values are weakly correlated with their optimal counterparts. In the AS models, the distributions of $|\Delta d_n|$ and $\sigma_{MOD}$ values are markedly different (see **Fig 1A and 1B**) and the average PCCs between them are 0.262, 0.277, 0.183 and 0.221 for the Cα-Cα, NO, SCMC and SCSC restraints groups respectively (see **Fig 4A**). Even with accurate alignments built through TM-align, the histogram-based approach of MODELLER produces $\sigma$ values which are weakly correlated to $|\Delta d_n|$ values (see **Fig 4B**).

In the previous section we have seen that the use of optimal $\sigma$ values greatly improves MODELLER's predictions. However, since $|\Delta d_n|$ values can not be directly inferred without the prior knowledge of the actual 3D structure that we are trying to predict, a strategy to improve MODELLER would consist in accurately estimating them. Irrespective of the predictive algorithm, it is reasonable to suppose that $|\Delta d_n|$ estimations will always bear a certain amount of error. In order to understand how 3D modeling quality changes as a function of this error, we rebuilt the models of the analysis set by perturbing their $|\Delta d_n|$ values with random noise.

**Effects on single-template modeling.** **Fig 5A** shows how the average GDT-HA of the AS models changes when increasing the fraction of $|\Delta d_n|$ values substituted with a random $\sigma$ (see **Fig 5B** for the relationship with lDDT). In the absence of any perturbation, the average GDT-HA is at its maximum of 0.6377. When the mean Cα-Cα $PCC_{MODEL}$ of the AS models is approximately 0.9, the average GDT-HA decreases by 2.6%. Further increasing the amount of random perturbation in $\sigma$ values leads to a continuous decrease in quality. When the average Cα-Cα $PCC_{MODEL}$ approximates 0, the average GDT-HA is 0.6056 (resulting in a 5.0% decrease with respect to the optimal state). This score is 0.8% higher than the average GDT-HA obtained using the default $\sigma_{MOD}$ values, which is 0.6009. Although the difference between these two scores is statistically significant (Wilcoxon signed-rank test, p-value = 1.6e-5) it is only minimal from a structural point of view. In other words, in single-template modeling, provided that the average $\sigma$ of a model does not surpass a certain threshold (that is, the average $|\Delta d_n|$ observed in nature), randomly generated $\sigma$ values are surprisingly as effective as those generated by the MODELLER histogram-based approach. This is also confirmed by the fact that the use of uniform $\sigma$ values $< 1.0$ Å does not significantly alter the GDT-HA and lDDT scores of models with respect to the standard MODELLER algorithm (see **Fig 2A and 2B**).

**Effects on multiple-template modeling.** Next, we performed perturbation experiments with multiple-template models (see **Fig 5C and 5D**). Again, the average quality decreases as perturbation increases. However, when the average Cα-Cα $PCC_{MODEL}$ approximates 0, the average GDT-HA now becomes 9.1% lower than the one obtained using the default MODELLER. This behavior is likely to be explained by the fact that in perturbation experiments the OL template weighting scheme was employed. When this scheme is applied with optimal (or near-optimal) $\sigma$ values, it boosts 3D modeling quality, but when it is applied with $\sigma$ values being weakly correlated with $|\Delta d_n|$ values, it has a detrimental effect (since for each HDDR it uses only the contribution of a randomly chosen template, while the contribution from the best template is likely to be suppressed).

This data shows that if we were able to predict $|\Delta d_n|$ values with sufficiently high accuracy, the performance of MODELLER would greatly increase. In single-template modeling, obtaining predictions with a PCC of ~0.6 would lead to an increase in GDT-HA of ~2.0%. In

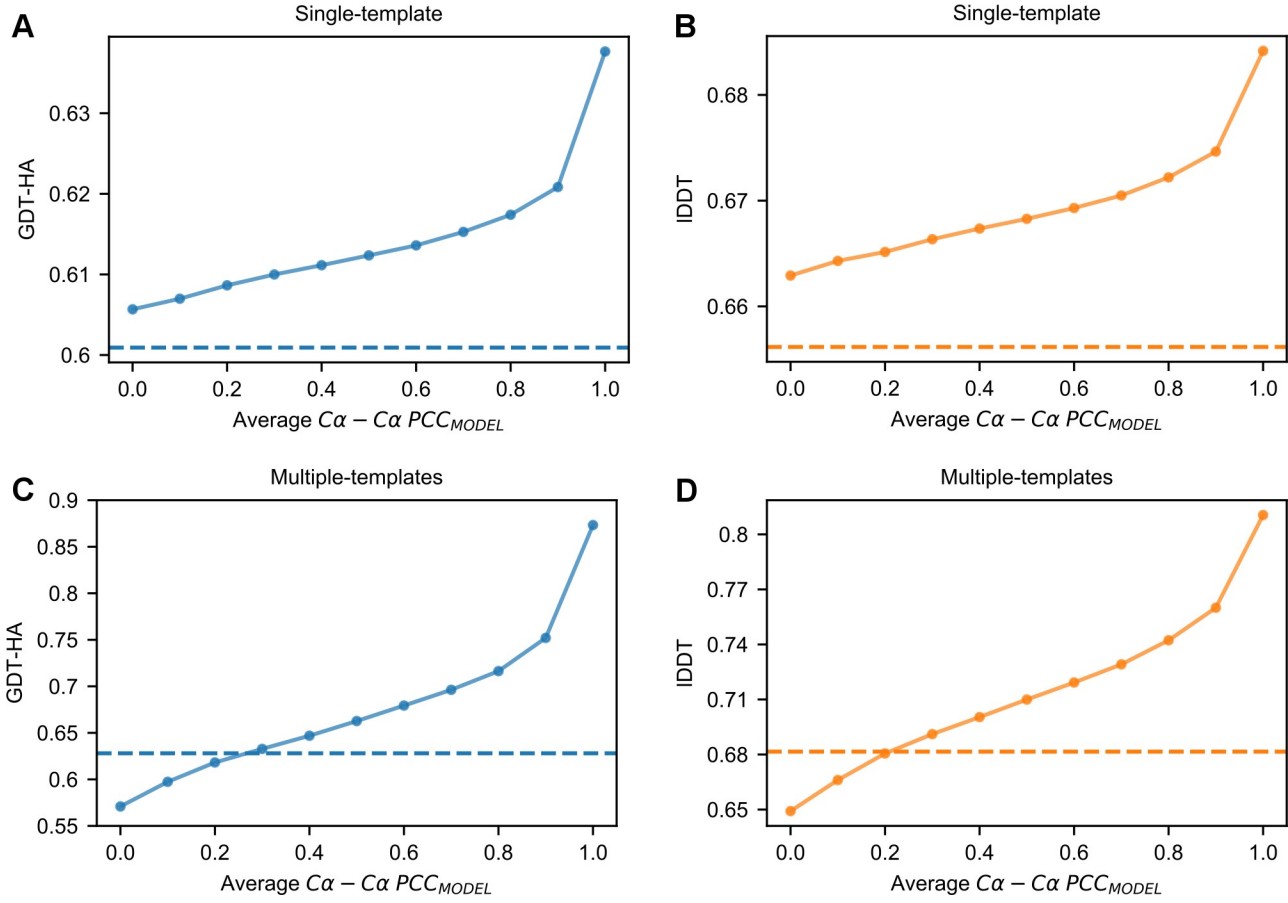

**Fig 5. Effect of $|\Delta d_n|$ perturbation on 3D modeling.** (A) and (B) Average GDT-HA and lDDT scores of the AS models as a function of their average Cα-Cα $PCC_{MODEL}$ values (see the "Methods" section). (C) and (D) Similar data obtained for the multiple-templates AM models. In (A) through (D), the dashed horizontal lines represent the average quality scores obtained by the default MODELLER.

multiple-template modeling, the potential gain is higher, as the same PCC would increase GDT-HA by a larger ~8.0%.

## Modifying the objective function of MODELLER with statistical potential terms

**Effect on single-template modeling.** In order to identify the optimal way to incorporate the DOPE potential within MODELLER, we performed benchmarks with the AS single-template models by tuning $w_{SP}$ values from 0.1 to 3.5 and by employing HDDRs bearing either $\sigma_{MOD}$ or $|\Delta d_n|$ values. **Fig 6A to 6C** show that, with both types of σ, the inclusion of DOPE leads to improvements in 3D modeling. Strikingly, depending on the type of σ, the amount of improvement and the best $w_{SP}$ vary greatly.

With $\sigma_{MOD}$ values, the maximum increase in GDT-HA is observed with a $w_{SP}$ of 0.5. As shown in **Table 3**, when employing DOPE with this $w_{SP}$, the average GDT-HA improves by a statistically significant 1.3% with respect to the default MODELLER. At the same time, the average lDDT score increases by 2.0%, showing that the use of DOPE also aids local modeling. Of note, when applying DOPE along with the *slow* MDSA protocol, an additional improvement is obtained: the average GDT-HA and lDDT scores now increase by 1.6% and 2.8%.

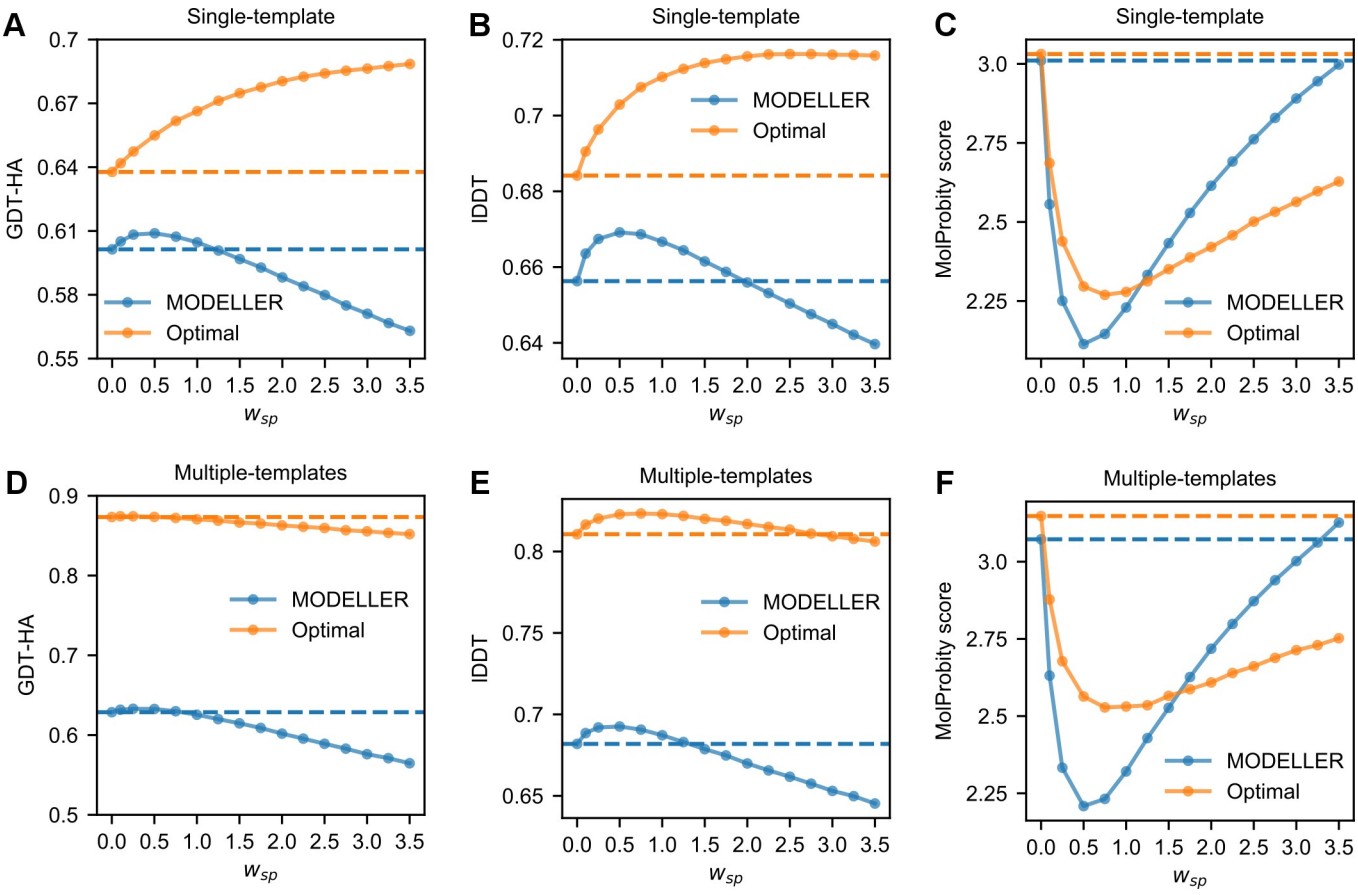

**Fig 6. Average quality scores of the models of the analysis set as a function of the $w_{SP}$ with which the DOPE potential has been included in the objective function of MODELLER.** (A) to (C) Quality scores of the AS models. (D) to (F) Quality scores of the AM models. (A) through (F) The horizontal dashed lines correspond to the scores obtained when modeling with MODELLER-generated (blue color) or optimal (orange) HDDRs without the use of DOPE.

When modeling with $|\Delta d_n|$ values, the best results are instead obtained with a $w_{SP}$ of 3.5. In this case, DOPE increases the average GDT-HA and lDDT scores by 8.0% and 4.6% with respect to the scores obtained with the same restraints and the standard objective function of

**Table 3. 3D modeling qualities of the AS single-template models built by including DOPE in the objective function of MODELLER.**

| Strategy | GDT-HA | lDDT | MolProbity score |
|---|---|---|---|
| MODELLER | 0.6014 (-) | 0.6563 (-) | 3.0104 (-) |
| OPTIMAL | 0.6377 (+6.0%)* | 0.6842 (+4.2%)* | 3.0311 (+0.7%)* |
| MODELLER-DOPE-0.5[a] | 0.6089 (+1.3%)* | 0.6692 (+2.0%)* | 2.1138 (-29.8%)* |
| MODELLER-SLOW-DOPE-0.5 | 0.6112 (+1.6%)* | 0.6746 (+2.8%)* | 2.0344 (-32.4%)* |
| MODELLER-DOPE-3.5 | 0.5631 (-6.4%)* | 0.6397 (-2.5%)* | 2.9977 (-0.4%) |
| OPTIMAL-DOPE-0.5 | 0.6549 (+8.9%)* | 0.7029 (+7.1%)* | 2.2960 (-23.7%)* |
| OPTIMAL-DOPE-3.5 | 0.6885 (+14.5%)* | 0.7158 (+9.1%)* | 2.6280 (-12.7%) |

See **Table 1** for the description of contents, columns and most modeling strategies names.

[a]The "DOPE-X.X" suffix indicates the use of DOPE with a $w_{SP}$ of X.X.

*Asterisks denote a statistically significant difference (according to a Wilcoxon signed-rank test with a significance level of 0.05) between the scores of a strategy and the scores of the default MODELLER. See **S3 Table** for a full list of the numerical p-values.

MODELLER. The increments in these two metrics are extremely large if computed with respect to the default MODELLER protocol (14.5% and 9.1%). **Fig 7** shows that with the default MODELLER, secondary structure elements that show divergence in the target and template structures are most often modeled in the template conformation. By using optimal HDDRs and DOPE, it is common to see these elements shifting towards target conformations.

Remarkably, the same $w_{SP}$ of 3.5 leads to a large decrease in modeling quality when DOPE is applied along with $\sigma_{MOD}$ values: in this case, the average GDT-HA and lDDT scores decrease by a large 6.4% and 2.5% with respect to the score obtained without using DOPE.

This data shows that in single-template modeling, the addition of DOPE is much more effective with $|\Delta d_n|$ values than with $\sigma_{MOD}$ values. Additional insights into this behaviour were provided by the analysis of DOPE energetic landscapes. **Fig 8** shows the representative case of the *1lam_chain_A* and *1dk8_chain_A* targets, where the DOPE energies of models are plotted as a function of their GDT-HA scores. When using single-template HDDRs with $\sigma_{MOD}$ values, applying DOPE with increasingly high $w_{SP}$ values leads to a decrease in both GDT-HA and DOPE energies. These energies eventually become even lower than the native target structure one. It seems that in the DOPE landscape, near-native conformations are not at an absolute minimum. On the other hand, when modeling with single-template optimal HDDRs, increasing $w_{SP}$ values leads to improvements in GDT-HA while maintaining DOPE energies relatively high. Similar trends are observed in the landscapes of almost all AS models. We speculate that this behaviour is caused by the fact that optimal HDDRs strongly restrain those regions of models which are structurally conserved between the native structures and templates, while they weakly restrain divergent regions. This probably allows to pinpoint the effect of DOPE in the divergent regions (where its addition likely improves modeling over the use of the standard MODELLER objective function) and to keep "rigid" the conserved regions (which are already extremely well-modeled and where DOPE can hardly improve the situation), thus giving rise to a synergistic effect.

**Effect on multiple-template modeling.**   Next, we explored the effect of DOPE in multiple-template modeling (see **Fig 6D** to **6F**). The trend observed when employing MODELLER-generated restraints is reminiscent of the single-template modeling one, although the improvements are slightly smaller. **Table 4** shows that the best $w_{SP}$ is 0.5, which results in an average increase in GDT-HA and lDDT of 0.6% and 1.6% with respect to the scores obtained with the default MODELLER. By employing DOPE with this $w_{SP}$ along with the *slow* MDSA protocol, an additional improvement can be reached: the average GDT-HA and lDDT scores now improve by 1.0% and 2.2%. When further increasing $w_{SP}$, we assist to a decrease in 3D modeling qualities.

The results observed when combining DOPE with optimal multiple-template HDDRs are different. No value of $w_{SP}$ is able to bring a relevant improvement in GDT-HA. As $w_{SP}$ increases over 1.0, the scores even start to decrease in a significant way, although it seems that DOPE is able to bring at least a small improvement in lDDT.

This counterintuitive behaviour can in part be explained from the analysis of DOPE energy landscapes. **Fig 8** shows that when using optimal multiple-template HDDRs, the quality of models is already higher than the one obtained with optimal single-template HDDRs. In this case, applying large $w_{SP}$ values leads to a decrease in DOPE energies and GDT-HA. The plots show that the models built with optimal HDDRs seem to be attracted towards a local energy minimum of DOPE, which does not correspond to the native state, but is located relatively near it. Therefore, when using optimal restraints, minimizing the DOPE of a structure distant from the native state (like in the case of single-template modeling), tends to increase its GDT-HA, but when the structure is already very close to the native state (such as in the case of multiple-template modeling), it tends to decrease its GDT-HA.

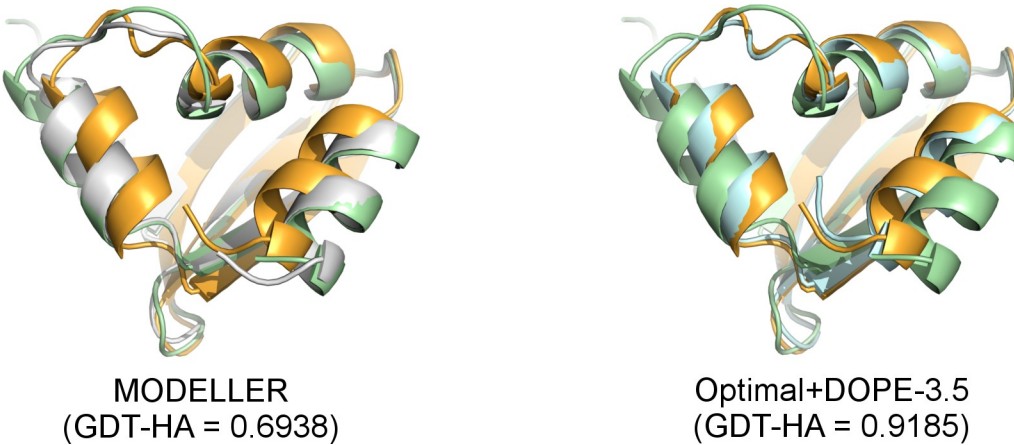

MODELLER
(GDT-HA = 0.6938)

Optimal+DOPE-3.5
(GDT-HA = 0.9185)

**Fig 7. Effects on 3D modeling of optimal $\sigma$ values and DOPE.** Effects brought by the use $|\Delta d_n|$ values and DOPE (with a $w_{SP}$ of 3.5) on the 3D modeling of target *1yd0_chain_A* (colored in orange) using as a template *1yd6_chain_D* (pale green). In the model built using the default MODELLER (colored in white, superposed to its target and template on the left image) the three helices shown in the image are positioned in the same conformation of the template. In the model built employing $|\Delta d_n|$ values and DOPE with a $w_{SP}$ of 3.5 (pale cyan, shown on the right) the helices are repositioned in a native-like conformation. Figures rendered with PyMOL [39].

**Effects on stereochemical quality.** In terms of stereochemichal quality, the use of DOPE seems to be highly beneficial in both single and multiple-template modeling and with both MODELLER-generated and optimal HDDRs (see **Fig 6, Tables 3** and **4**). For example, when employing $\sigma_{MOD}$ values and DOPE with a $w_{SP}$ of 0.5, the average MolProbity score of the AS models decreases by a large 29.8% with respect to the default MODELLER. Additional improvements in MolProbity scores are observed when coupling DOPE to the *slow* MDSA protocol. We found that the MolProbity score component in which DOPE brings the largest improvement is by far the "Clash Score", meaning that the potential helps to remove steric

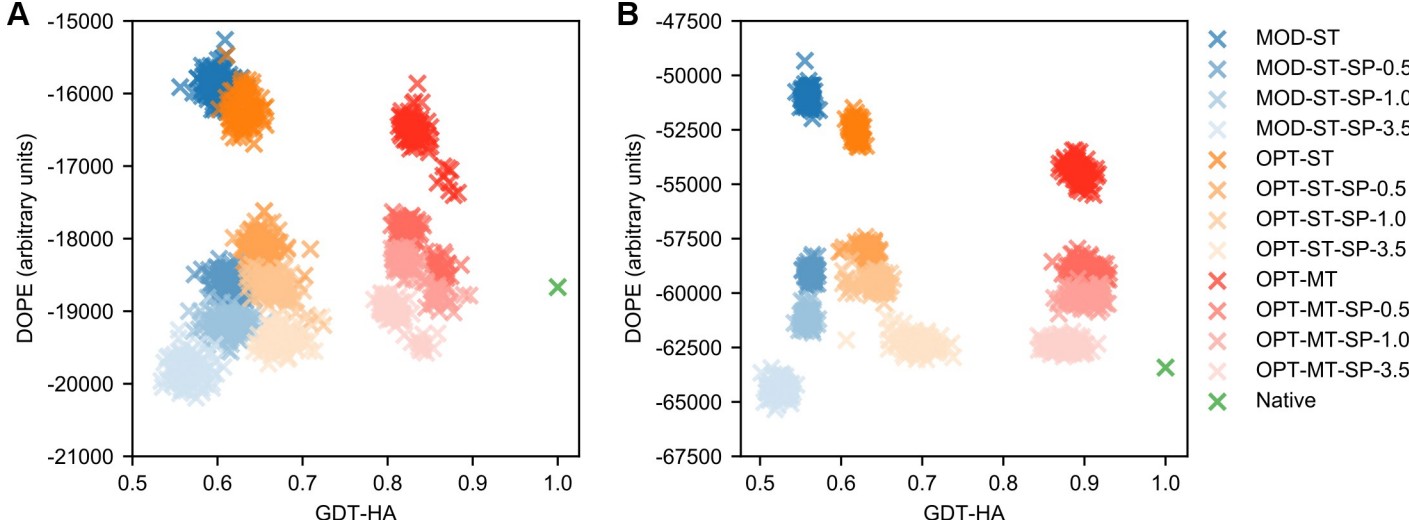

**Fig 8. DOPE energy landscapes.** DOPE energy landscapes for target (A) *1dk8_chain_A* and (B) *1lam_chain_A* modeled using different strategies. 100 decoys were built for each strategy and their GDT-HA scores are plotted here against their DOPE energies. The strategies with the "MOD-ST" prefix adopted MODELLER-generated HDDRs and a single template (blue-shaded dots), those with the "OPT-ST" prefix adopted optimal HDDRs and a single template (orange-shaded dots) and those with the "OPT-MT" prefix adopted optimal HDDRs and multiple templates (red-shaded dots). The "SP-X.X" suffix indicates the use of DOPE with a $w_{SP}$ of X.X. The green dots correspond to the DOPE-minimized native target structure.

**Table 4. 3D modeling qualities of the AM multiple-template models built by including DOPE in the objective function of MODELLER.**

| Strategy | GDT-HA | lDDT | MolProbity score |
|---|---|---|---|
| MODELLER | 0.6287 (-) | 0.6819 (-) | 3.0725 (-) |
| OPTIMAL | 0.8733 (+38.9%)* | 0.8106 (+18.9%)* | 3.1478 (+2.4%) |
| MODELLER-DOPE-0.5 | 0.6327 (+0.6%)* | 0.6926 (+1.6%)* | 2.2086 (-28.1%)* |
| MODELLER-SLOW-DOPE-0.5 | 0.6347 (+1.0%)* | 0.6971 (+2.2%)* | 2.1152 (-31.2%)* |
| MODELLER-DOPE-3.5 | 0.5646 (-10.2%)* | 0.6453 (-5.4%)* | 3.1267 (+1.8%)* |
| OPTIMAL-DOPE-0.5 | 0.8736 (+39.0%)* | 0.8229 (+20.7%)* | 2.5635 (-16.6%)* |
| OPTIMAL-DOPE-3.5 | 0.8519 (+35.5%)* | 0.8061 (+18.2%)* | 2.7520 (-10.4%)* |

See **Table 1** for the description of contents, columns and most modeling strategies names.

*Asterisks denote a statistically significant difference (according to a Wilcoxon signed-rank test with a significance level of 0.05) between the scores of a strategy and the scores of the default MODELLER. See **S4 Table** for a full list of the numerical p-values.

clashes from models. Therefore, the inclusion of DOPE in the objective function of MODELLER represents a fast and effective way of improving the stereochemical quality of its models. This approach increases computational times by a factor of ~6.5 when employing the *very_fast* MDSA protocol (and ~16.5 with the *slow* protocol), but on modern hardware the default MODELLER algorithm usually takes a few seconds to complete a model, therefore in absolute terms the model building process is still relatively fast.

**Comparison between DOPE and DFIRE in 3D modeling.** We also tested the effect of adding DFIRE in the objective function of MODELLER. Overall, DFIRE seems to have very similar effects to the ones described for DOPE (see **S3 Table**, **S4 Table** and **S5 Fig**), because their terms have very similar forms (see **Fig B** in **S3 Fig**). However, when modeling with $\sigma_{MOD}$ values, DOPE seems to slightly outperform DFIRE in terms of all-atom local quality (expressed by lDDT scores). When using a $w_{SP}$ of 0.5 and $\sigma_{MOD}$ values, DOPE yields for the AS models an average lDDT score 0.5% higher than the one obtained with DFIRE, a small but statistically significant improvement (Wilcoxon signed-rank test, p-value = 4.6e-35). Therefore, we suggest that in MODELLER, DOPE should be preferred over DFIRE.

## Discussion

Improving the quality of HM predictions is clearly an area of great relevance in Biomedical Research [40], given that the applicability of this methodology is expected to increase in the next years [5]. Right now, a large portion of targets can be modeled only with low accuracy, due to the remote homology relationship (under 30% SeqId) with their templates. A solution to this problem could potentially come from advances in 3D model building or refinement algorithms. In this work, we have explored two main promising strategies to increase the accuracy of the original MODELLER algorithm.

The use of optimal $\sigma$ values (that is, $|\Delta d_n|$ values) greatly increases the 3D modeling quality of the program. Since $|\Delta d_n|$ values can only be obtained by knowing the exact amount of divergence between the structure of a target and its templates, they can not be used in real-life protein structure prediction scenarios (where the target structure is of course unknown).

However, as first shown by the Lee group [22], $|\Delta d_n|$ values may be estimated through a machine learning system. These authors developed a random forest which obtained estimations with an average Cα-Cα PCC of ~0.35. The use of this predictor led to only a very small improvement in terms of 3D modeling quality. Our data (which describes the relationship between 3D modeling quality and errors in $|\Delta d_n|$ estimations) shows that increasing the PCC

of a similar predictor by at least 0.2–0.3 units could translate in a significant improvement of MODELLER.

The other strategy that we have investigated is the inclusion of statistical potential terms, such as DOPE, in the objective function of MODELLER. We show that employing such potentials in the 3D model building phase of MODELLER robustly increases 3D modeling quality and provides a fast and effective way to improve the stereochemical details models. In order to allow the user community of MODELLER to deploy this strategy in their modeling pipelines, we share the Python code implementing it. In future research, it will be interesting to see if there exist potentials with an even more beneficial effect on 3D model building in MODELLER.

Our results have implications also for other Structural Bioinformatics tools. RosettaCM and I-TASSER borrow from MODELLER the use of HDDRs [36, 41–42] and programs like MUL-TICOM [43] and Pcons [44] implement MODELLER at some point in their protein modeling pipelines. The strategies presented in this work can certainly be implemented in these protocols to improve their quality.

Of note, in the protein structure refinement field, restraints are built from a starting model and the aim is to guide the model towards its native conformation [45]. While in the HM context we may estimate $|\Delta d_n|$ values between a target native structure and a template, in protein structure refinement they could be similarly estimated between a native structure and its unrefined model. Methods to predict the local accuracy of 3D models already reach good performances [46]. It is reasonable to think that with a sufficiently accurate predictor, the $|\Delta d_n|$ prediction strategy could also lead to improvements in current refinement strategies.

The development of deep learning techniques [47] has recently brought advances in the field of contact and distance map prediction [48]. We suggest that such methodologies could be well adapted to the problem of $|\Delta d_n|$ estimation. In future studies, we will concentrate on using this type approach to tackle the problem of $\sigma$ values assignment. Since a machine learning model usually performs predictions in a relatively small amount of time, the $|\Delta d_n|$ estimation approach has the potential to greatly improve the "modeling by satisfaction of spatial restraints" strategy of MODELLER at the price of small computational cost.

## Supporting information

**S1 Table. Physical terms of the MODELLER objective function.** Note how by default the objective function does not include any "physical" attractive term between non-bonded atoms (Lennard-Jones and Coulomb potential terms from CHARMM22 [1] are missing). The only attractive terms in the objective function are homology-derived distance restraints (see **S2 Table**).
(PDF)

**S2 Table. Homology-derived terms of the MODELLER objective function.**
(PDF)

**S3 Table. 3D modeling qualities of the AS single-template models built with different modeling strategies.** See **Table 1** in the main text for the description of contents, columns and most modeling strategies names.
(PDF)

**S4 Table. 3D modeling qualities of the AM multiple-templates models built with different modeling strategies.** See **Tables 1** and **2** in the main text and **S3 Table** for the description of contents, columns and most modeling strategies names.
(PDF)

**S1 Fig. Properties of the analysis set.** (A) SeqId histogram of the pairwise target-template alignments in the AS models obtained using TM-align and HHalign. (B) Target coverage histograms of the same alignments. (C) Chain length histograms of the 225 AS targets, the 118 AM targets and all the 472 template chains of the analysis set. (D) CATH classes frequencies of the AS and AM targets compared to those in the entire CATH 4.2.0 database [1].
(PDF)

**S2 Fig. Details of the $|\Delta d_n|$ perturbation scheme.** (A) to (E) the Cα-Cα $|\Delta d_n|$ values of the *5jwo_chain_B* (target) - *1thx_chain_A* (template) pair were perturbed to various $PCC_{SEL}$ levels using the perturbation scheme described in the "Methods" section of the main text. The observed PCCs between the perturbed and the original $|\Delta d_n|$ values are reported. (F) Distributions of the original $|\Delta d_n|$ values and three perturbed values lists shown in previous figures. The mean values of the lists are reported in brackets. Thanks to the use of Laplace distributions for extracting random errors, the perturbed values are distributed approximately as exponentials, which resemble the original $|\Delta d_n|$ distribution. (G) and (H) average $PCC_{MODEL}$ values of the AS and AM models in $|\Delta d_n|$ perturbation experiments plotted as a function of $PCC_{SEL}$. On average, each $PCC_{SEL}$ value allows to obtain almost exactly the desired level of perturbation (quantified as $PCC_{MODEL}$). Data for the four HDDRs groups of MODELLER is shown. (G) AS models. (H) AM models.
(PDF)

**S3 Fig. Analysis of the terms of the DOPE and DFIRE potentials.** (A) Forms of the 12561 terms of DOPE [1]. Each term is associated to a couple of heavy atom types from the 20 standard residues. Irrespective of the atom types, all the functions start to acquire a flat shape above the 8.0 Å threshold. (B) Confrontation of DOPE and DFIRE [2] terms. An hexbin density plot compares 364269 data points from all the 12561 terms of DOPE (x-axis) and DFIRE (y-axis) (each term has 29 points, which report the score of the potential in a linear space from 0.75 to 14.75 Å). The scores of the two potentials are highly correlated (Pearson correlation coefficient = 0.99).
(PDF)

**S4 Fig. Accuracy of the pairwise target-template HHalign alignments of the AS models.** The x-axis reports the SeqId between the target and template sequences in TM-align alignments. The y-axis reports the accuracy of the corresponding HHalign alignment. The accuracy is computed as the ratio $H_m/T_m$, where $T_m$ is the total number of matches in the TM-align alignment and $H_m$ is the number of "correct" matches in HHalign alingments (that is, those HHalign matches which are also found in the TM-align alignment). The average accuracy is 0.87.
(PDF)

**S5 Fig. Average quality scores of the analysis set models as a function of the $w_{sp}$ value with which the DFIRE or DOPE statistical potentials have been included in the objective function of MODELLER.** The horizontal dashed lines correspond to the scores obtained when modeling with MODELLER-generated (blue color) or optimal (orange) HDDRs without the use of statistical potentials. (A) to (C) quality scores of the AS models. (D) to (F) quality scores of the AM models.
(PDF)

**S1 Text. Description of the GDT-HA and lDDT metrics for model quality evaluation.**
(PDF)

**S2 Text. Obtaining optimal parameters for single-template HDDRs.**
(PDF)

**S3 Text. Obtaining optimal parameters for multiple-template HDDRs.**
(PDF)

**S4 Text. Distribution of best templates for multiple-template HDDRs along target sequences.**
(PDF)

## Acknowledgments

The authors wish to acknowledge Fabio Mastrantuono and Fransceso Pesce for helpful discussions and their precious help.

This work is dedicated to the memory of our beloved mentor Prof. Francesco Bossa.

## Author Contributions

**Conceptualization:** Giacomo Janson, Alessandro Paiardini.

**Data curation:** Giacomo Janson, Alessandro Grottesi.

**Formal analysis:** Giacomo Janson.

**Funding acquisition:** Alessandro Paiardini.

**Investigation:** Giacomo Janson, Marco Pietrosanto.

**Methodology:** Giacomo Janson, Marco Pietrosanto.

**Project administration:** Alessandro Paiardini.

**Resources:** Alessandro Grottesi, Alessandro Paiardini.

**Software:** Giacomo Janson, Alessandro Grottesi, Marco Pietrosanto.

**Supervision:** Gabriele Ausiello, Giulia Guarguaglini, Alessandro Paiardini.

**Validation:** Giacomo Janson.

**Writing – original draft:** Giacomo Janson.

**Writing – review & editing:** Alessandro Grottesi, Gabriele Ausiello, Giulia Guarguaglini, Alessandro Paiardini.

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
