## [Decision Letter · Decision Letter 0]

1 Oct 2019

Dear Dr JANSON,

Thank you very much for submitting your manuscript 'Revisiting the “satisfaction of spatial restraints” approach of MODELLER for protein homology modeling' for review by PLOS Computational Biology. Your manuscript has been fully evaluated by the PLOS Computational Biology editorial team and in this case also by independent peer reviewers. The reviewers appreciated the attention to an important problem, but raised some substantial concerns about the manuscript as it currently stands. While your manuscript cannot be accepted in its present form, we are willing to consider a revised version in which the issues raised by both reviewers have been adequately addressed. We cannot, of course, promise publication at that time.

Sincerely,

Bert L. de Groot

Associate Editor

PLOS Computational Biology

Arne Elofsson

Deputy Editor

PLOS Computational Biology

[LINK]

Reviewer's Responses to Questions

**Comments to the Authors:**

Reviewer #1: The manuscript describes an analysis of the potential to improve homology modelling using “satisfaction of spatial restraints” in the widely used MODELLER package. If the structural divergence between target and template is optimally modelled there is a large room for improvement in particular for multiple templates. Improvements (2% for single and 11% for multiple templates) are expected if the predicted structural divergence correlate >0.6 compared to the true divergence. In addition, the authors also investigate the possibility to include a statistical potential in the objective function of the MODELLER and show that using the build in DOPE statistical potential yields a small but consistent improvement in model quality. Code to for the latter is provided in a git repo.

Overall this manuscript was a nice read and present results that can be used to consistently improve modelling results. It also establish an upper limit on model quality that can be gained if the templates are used optimally. However, if I should provide some criticism, a large part of the analysis is done using information from the native structure when setting the local weights. If you know exactly which residues to move and to which degree, you should expect large improvements.

1) The authors provide some perturbation analysis by randomly changing some fraction of weights from their optimal value, thereby reducing the correlation from 1.0 to 0.0 and show that improvements are expected when the correlation is >0.6 (Fig 5). Looking at Fig S2, 0.6 correlation corresponds to changing only 30% (0.3) of the residues. Thus, 70% of the residues have their structural divergence at their optimal and 30% are random. This scenario is highly unrealistic; 0.6 correlation doesn’t seem too high, but since correlation is really effected by small number of random points buried among perfect predictions, the actual required prediction performance might be much higher. I suggest that the perturbation analysis is performed in a more rigorous way that sample distributions more likely to originate from a model quality assessment prediction method, where each estimate would have some uncertainty, Or better use a proper model quality assessment program, like ProQ3D or QMEAN to get realistic estimates.

2) Best performance gain can potentially be obtained using multiple templates. However, again here, the fact the authors are using the knowledge on which template is optimal obfuscates the true value of this results. We already know that if you are able to always pick the best model you would outperform any group in CASP, the problem is to pick the best model/template. Thus, it is crucial that effect of errors in the estimates are investigated more throughly.

3) For multiple templates how are the delta(d_ij) for different templates distributed through the target sequence? Is it different templates that dominates in different regions, or are they intertwined? i.e. is it effectively using more than one template for any given region or is it more picking the best template for each region?

4) It is a bit unclear on how the local weights are implemented, do you provide a custom made restraints file to Modeller? or did you find any other API to interface with the Modeller functions? Anyway I think it would be useful if you in addition to the code you already provide, also provide code for running Modeller with the optimal weight (if native is available), or user-specified given a list of local predicted CA-CA distances.

Reviewer #2: This manuscript describes a study of MODELLER, a widely-used software tool for protein homology modeling.

Given that MODELLER is widely used, further study of this program and even a small improvement are always desirable.

In this manuscript, the authors have studied the relationship between modeling quality and estimation of the difference of an inter-atom distance between target and template. The authors claim that 1) a more accurate estimation of the difference may improve modeling accuracy and 2) modeling quality may be increased by incorporating some statistical potentials into MODELLER. These findings are not very new, but the manuscript provides sufficient data and analysis to back up them, which to the best of my knowledge is not widely available in the literature. The authors have also released source code for the incorporation of DOPE and DFIRE into MODELLER, although this is not very new (Similar code is available at the MODELLER website).

Some minor concerns:

1) lines 46-47, it is fine to say that template-based modeling is the most popular, but I am not sure if it is fine to claim that the most successful approach is template-based modeling since this method fails on the modeling of many membrane proteins. In particular, in the past 2-3 years template-free modeling has made a very good progress and now its accuracy is comparable or even better as long as the target protein does not have very good templates. Further, currently the best template-free modeling also works well on membrane proteins.

2) lines 69-71, some revision is needed here. In addition to algorithm advance, the enlargement of both sequence and structure databases is also a very important factor for the improvement of HM modeling.

**Have all data underlying the figures and results presented in the manuscript been provided?**

Reviewer #1: Yes

Reviewer #2: Yes

PLOS authors have the option to publish the peer review history of their article (what does this mean?). If published, this will include your full peer review and any attached files.

Reviewer #1: Yes: Björn Wallner

Reviewer #2: No

---

## [Decision Letter · Decision Letter 1]

13 Nov 2019

Dear Dr JANSON,

We are pleased to inform you that your manuscript 'Revisiting the “satisfaction of spatial restraints” approach of MODELLER for protein homology modeling' has been provisionally accepted for publication in PLOS Computational Biology.

In the meantime, please log into Editorial Manager at https://www.editorialmanager.com/pcompbiol/, click the "Update My Information" link at the top of the page, and update your user information to ensure an efficient production and billing process.

One of the goals of PLOS is to make science accessible to educators and the public. PLOS staff issue occasional press releases and make early versions of PLOS Computational Biology articles available to science writers and journalists. PLOS staff also collaborate with Communication and Public Information Offices and would be happy to work with the relevant people at your institution or funding agency. If your institution or funding agency is interested in promoting your findings, please ask them to coordinate their releases with PLOS (contact ploscompbiol@plos.org).

Thank you again for supporting Open Access publishing. We look forward to publishing your paper in PLOS Computational Biology.

Sincerely,

Bert L. de Groot

Associate Editor

PLOS Computational Biology

Arne Elofsson

Deputy Editor

PLOS Computational Biology

Reviewer's Responses to Questions

Comments to the Authors:

Please note here if the review is uploaded as an attachment.

Reviewer #1: All my comments have been adequately addressed.

Have all data underlying the figures and results presented in the manuscript been provided?

Large-scale datasets should be made available via a public repository as described in the 

PLOS Computational Biology

data availability policy, and numerical data that underlies graphs or summary statistics should be provided in spreadsheet form as supporting information.

Reviewer #1: Yes

PLOS authors have the option to publish the peer review history of their article (what does this mean?). If published, this will include your full peer review and any attached files.

Do you want your identity to be public for this peer review?

 For information about this choice, including consent withdrawal, please see our Privacy Policy.

Reviewer #1: Yes: Björn Wallner

---

## [Editor Report · Acceptance letter]

10 Dec 2019

PCOMPBIOL-D-19-01048R1 

Revisiting the “satisfaction of spatial restraints” approach of MODELLER for protein homology modeling

Dear Dr JANSON,

I am pleased to inform you that your manuscript has been formally accepted for publication in PLOS Computational Biology. Your manuscript is now with our production department and you will be notified of the publication date in due course.

With kind regards,

Laura Mallard
